# Influence of Wet Ageing on Beef Quality Traits

**DOI:** 10.3390/ani13010058

**Published:** 2022-12-23

**Authors:** Francesco Sirtori, Silvia Parrini, Maria Chiara Fabbri, Chiara Aquilani, Aldo Dal Prà, Alessandro Crovetti, Giovanni Brajon, Riccardo Bozzi

**Affiliations:** 1Department of Agriculture, Food, Environment and Forestry (DAGRI), University of Florence, 50144 Florence, Italy; 2Institute of Bioeconomy (IBE), National Research Council, 50145 Florence, Italy; 3Istituto Zooprofilattico Sperimentale del Lazio e Della Toscana “M. Aleandri”, Via Castelpulci 43, 50018 Florence, Italy

**Keywords:** wet ageing, beef cattle, meat quality, meat storage, Longissimus thoracis

## Abstract

**Simple Summary:**

The modern consumer is increasingly attentive to the characteristics of products, including meat, both from an organoleptic point of view and from that of food safety. At the same time, the need to reduce costs associated with product storage is increasingly pressing. In this framework, it is therefore important to study the qualitative response of the meat product subjected to different ageing times using the wet method. Finding the right time to age meat will allow the obtention of higher meat quality for consumers while also reducing storage costs. Therefore, this study tries to analyze the influence of the storage technique on the meat quality, taking into consideration other fundamental factors present in the marketed meat, such as, first of all, the breed, age and the potential animal stress.

**Abstract:**

Fresh samples of *Longissimus thoracis* of Charolais (*n* = 12), Romagnola (*n* = 15), Limousine (*n* = 77), and crossbreed (*n* = 62) animals were evaluated with different storage periods (0, 4, 9, and 14 days). Proximate analysis (i.e., pH, humidity, color, free water content, and physical parameters) was performed for each sample. The data obtained were evaluated with a mixed model, setting 5 fixed effects (breed, storage time, animals age, EUROP conformation, number of animal transports) and the animal as random. The results demonstrated that meat quality was affected mainly by the wet ageing period and that the visual and tactile parameters were also found to be susceptible to the storage time. The conservation entailed a decrease in meat humidity and an increase in L* and b* traits; it also led to a decrease in the hardness of the sample, in turn affecting the other texture profile analysis parameters considered. Fixed effects affected in different ways the traits analyzed i.e., pH and humidity changed with breed, as well as with EUROP classification, animals’ age for some TPA parameters, and the number of animal transports for both visual and tactile parameters. Wet ageing influenced the meat quality, often improving it, confirming how important further research would be to identify precise storage times in relation to the parameters studied.

## 1. Introduction

Global livestock production in the world has increased substantially since the 1960s, and beef production has more than doubled. This because a growing demand associated with human population income and urbanization growth and the consequent fast lifestyles changes in both diet and physical activity [1,2]. Population growth is clearly the main driver of increased consumption, and the projected global increase of 11% will underpin a projected increase of 14% in global meat consumption by 2030 [3]. The determinants of meat consumption are complex. Demography, urbanization, incomes, prices, traditions, religious beliefs, cultural norms, environmental, ethical/animal welfare, and health concerns are key factors that surely affect the amount but also the type of meat consumption. Diets in industrialized countries are characterized by high the consumption of animal products, with a continuous pursuit of a higher healthiness, technological, and organoleptic qualities. This is reinforced by continuous recommendations from national and international organizations, such as the World Health Organization (WHO) and World Cancer Research Fund, which suggest no more than 300–400 g of red meat per week [4,5]. Ageing produces necessary changes in meat characteristics, and the optimum ageing time depends, among other factors, on the breed and the age at slaughter [6]. An increasing consumer demand for a higher quality and eating experience has led to revisiting the dry ageing process [7]. The maturing process may causes changes in sensory attributes [8], technological parameters [9], color [10], and pH [11]. Two ageing methods are usually applied to beef: i) dry ageing, wherein carcasses or primal cuts are hung in open air at controlled temperature, usually for two–four weeks; ii) ageing in vacuum packs, widely used to preserve and to improve the meat quality [12]. The first method results in gradual dehydration, thus decreasing yield due to the weight loss and due to the need for surface trimming [9], and is a common process to improve meat quality characteristics (e.g., tenderness and texture). The second method, wet ageing, is a process introduced in the 1970s, and vacuum packaging is used to protect the meat from spoiling and drying out when stored for ageing in a refrigerated environment between 3 and 83 days [7]. Wet ageing is also the most common packaging in the meat industry because most palatability attributes of meat are improved with this process. With wet ageing, quarters or primal cuts are vacuum-sealed in plastic bags and kept for 4–20 days to obtain acceptable tenderness [9], which represents one of the most important characteristics considered by consumers, determining the decision for repeat purchase [13]. Wet ageing also results in a lower cost for the purchaser [14]. The aim of this study was to analyze the effects of wet ageing at different days (1, 4, 9, and 14) on physical–chemical beef quality traits and how these are affected by various factors (e.g., breed, age).

## 2. Materials and Methods

### 2.1. Meat Samples Collection

The meat samples came from different genetic types: 77 Limousine, 62 crossbred (dairy cow x beef bull), 15 Romagnola, and 12 Charolais male animals pasture raised (with an indoor finishing feeding period) and with an age between 12 and 24 months and collected from a commercial beef slaughter facility located in Scarperia, Tuscany (Italy). Following the slaughter procedure, the carcasses were graded using the EU Beef Carcass Classification Scheme with an average hot carcass weight of 231 kg and ultimate pH of 5.65. Twenty-four hours after slaughtering, the carcasses were dissected and 4 steaks containing the muscle *Longissimus thoracis* (LT) were taken from the left half-carcass of each animal at the level of the 6th and 10th thoracic vertebra for a total of 824 identified meat samples. Steaks were transported under refrigeration and kept at 4 °C until the analysis.

### 2.2. Wet Ageing Specification

The 4 steaks from each animal were dissected, and the LT muscle sample of a steak was used as a reference sample at time “zero” (T0), while the other 3 LT muscles were wet-aged in vacuum bags at one of the predetermined storage times: 4, 9, and 14 days at 4 °C (T4, T9, and T14, respectively).

### 2.3. Chemical and Physical Meat Analysis

After the preparation of LT muscle samples, the following chemical analyses were carried out on fresh meat of each animal classified as T0: humidity, crude protein, ash [15], and intramuscular fat as total lipid content using a modified method of Folch et al. (1957) [16]. The total lipid was extracted from 2 g of sample with 37 mL chloroform–methanol, 2:1 (*v/v*) containing 0.01% of BHT. The lipid extract was washed by adding 10 mL of 0.88% KCl. The surnatant was recovered, and the solvent removed. The purified lipid extract was dissolved in chloroform (5 mL). The following physical analyses were performed on all samples (T0, T4, T9, and T14) produced from the 4 steaks of each animal. The final pH after 24 h was recorded using a pH-meter Delta Ohm HD 8705 (Delta Ohm S.r.L., AOAC, 2000 Caselle di Selvazzano, Padova, Italy) with temperature probe TP870 and pH electrode Hamilton double pore and calibration performed automatically at pH 4.01 and 6.00. The color parameters CIE L* (lightness), a* (redness), and b* (yellowness) were determined using a Minolta colorimeter CR-200 (Minolta Camera Co., Ltd., Osaka, Japan) with a white and red calibration plate. The colorimeter was recalibrated with the two standards at the start of each measuring session. The top of the Chroma Meter measuring head was placed in a flat position against the meat’s surface. The reflective color was determined from the average of three consecutive pulses from the optical chamber of the colorimeter. For each sample, three measurements were performed. Data are reported in the L* a* b* color notation system (https://cie.co.at/publications/colorimetry-3rd-edition (accessed on 11 November 2022)), with the L* axis representing lightness, the a* axis representing the green–red color axis (redness), and the b* axis representing the blue–yellow (yellowness) color axis. Texture profile analysis (TPA) was performed using a Zwick Roell Z2.5 apparatus (Ulm, Germany texture analyzer) with a 1 kN-load cell at a crosshead speed of 1 mm/s and working at room temperature (22 °C). TPA curve-forces were determined by a 100 mm-diameter compression plate on a 10 × 10 × 10 mm slice, and a double compression cycle test was performed, using up to 25% compression of the original portion heights. Hardness (peak force of the first compression cycle), springiness (the distance of the detected height during the second compression divided by the original compression distance), and cohesiveness (ratio of positive area of the force during the second compression compared to that obtained during the first compression) were recorded. Chewiness was calculated as hardness multiplied by cohesiveness multiplied by springiness [17]. The free water has been assayed using the filter paper press method [18]. The descriptive statistics of each parameter analyzed was reported in Table 1.

### 2.4. Statistical Analysis

The correlation matrix between parameters analyzed was evaluated, and the plot was carried out using the corrplot R package [19]. Records for each parameter (i.e., pH, humidity, L*, a*, b*, free water, hardness, cohesiveness, chewiness, and springiness) were tested for normality and homoscedasticity and then analyzed with the following mixed linear model, using the R function lmer of lme4 R package [20]:
$$Y_{\mathrm{ijklmno}}=µ+{\;B}_{i}+{\;D}_{j}+{\;\mathrm{Age}}_{k}+{\;S}_{l}+{\;C}_{m}+{\;A}_{n}+{\;e}_{\mathrm{ijklmno}}$$
where Y is in turn the observation for pH, humidity, L, a, b, free water, hardness, cohesiveness, chewiness, and springiness; µ = the overall mean; B = fixed effect of the ith breed (4 levels: Charolais, Crossbred, Limousine, and Romagnola); D = fixed effect of the jth day of wet ageing (4 levels: 1, 4, 9, and 14 days); Age = fixed effect of the kth age class of slaughtered animals (4 levels: 1 (12–15 months), 2 (16–18 months), 3 (19–21 months), and 4 (22–24 months); S = fixed effect of the lth number of farms where bulls have passed through (4 levels: 0, 1, 2, and 3 transports); C = fixed effect of the mth EUROP class (4 levels: E, U, R, and O); A = random effect of the nth bull (166 levels); and e = random residual. Records of protein, lipid, and ash contents were tested with the previous mixed linear model without the duration-of-wet-ageing fixed effect, because parameters evaluated only on day 1. The least square means have been calculated with the R emmeans package [21] for both models.

## 3. Results

By analyzing the data obtained, it is possible to report the correlations among the parameters (Figure 1). As expected, the lipids percentage was inversely correlated to the percentage of humidity; in addition, the lipids were positively correlated with the b* value of the color. The colorimetric coordinates L* and b* were found to be positively correlated. The hardness had a correlation equal to 1 with the chewiness, as expected, because this latter is a ratio containing the former and springiness values.

As regards the chemical characterization of fresh LT (Table 2), the breed was the main factor of variability showing significant differences both for protein and intramuscular fat contents, highlighting the Limousine as the breed with the highest protein value and lowest fat content than the other groups (*p* < 0.01).

Least square means related to animal age and EUROP-conformation classes did not show significant differences among groups. Instead, significant differences in LT protein content have been highlighted in relation to the step factor (number of times the animal had been subjected to transport); however, it did not show a linear trend.

The results carried out at various storage periods were reported in Table 3.

The wet ageing seemed to influence all the parameters considered except for the red color (a*) and springiness. Almost all of the parameters showed a linear trend, positive or negative, with the increase in wet ageing days. Indeed, the variation in pH showed an increase in values from T4 (5.52) to T9 (5.56), while at T14, the pH value decreased to 5.54. A significant reduction in the humidity value was observed among the days from 74.13 (T0) until 73.29% (T14). The wet ageing meat samples resulted in a constant increase in the lightness (L*) and yellowness (b*). L* values varied from 41.09 (T0) to 42.18 at the 14th day of maturation, while b* values varied from 6.41 to 6.98 at T0 and T14, respectively. According to the free water analysis, there was a significant difference between the four wet ageing times: the lowest value was observed with the ageing time of 14 (0.54). As regards TPA parameters, the maturation of the meat led to a sample tenderization (from 1.81 (T0) to 1.03 kg/cm^2^ (T14)), which in turn influenced the other traits, such as cohesiveness (from 0.40 in T0 up to 0.46 in T14), and chewiness, which had undergone a decrease from 0.67 (T0) to 0.42 (T14).

As regards breed effect, pH values varied from 5.48 to 5.60 for Romagnola and Charolais, respectively, even if all values fell within the normal range of pH. Furthermore, humidity had different values among breeds, resulting in the highest value in Limousine. Animal age classes showed significant differences (*p* < 0.01) for two related parameters of TPA analysis: hardness and chewiness. In the intermediate classes (2 and 3), the highest values of both parameters were highlighted. The EUROP classes reported significant differences for pH and humidity, showing the lowest and the highest value in the E category (5.46 of pH and 74.83% of humidity). Another difference for this factor was found in hardness, wherein the E and R classes had higher values (1.46 kg/cm^2^) than the other classes. As regards the number of transports that animals have been subjected to, humidity, b* values, and hardness were the parameters that showed significant differences (*p* < 0.01) among levels. Humidity percentage resulted in a fluctuating increase in estimates, while the b* value decreased with the increase in the animal transports number (from 7.24 to 6.17). The hardness presented non-linear results, resulting in the highest values for classes 0 and 2.

## 4. Discussion

The present study aimed to evaluate the effect of wet ageing on chemical and physical meat characteristics. Among the numerous processes used for the product preservation, the common ones are dry ageing and wet ageing. To investigate how fundamental ageing time is, several authors have highlighted how the optimal maturation time is difficult to determine, but they all agreed it to be shorter in the process of wet ageing [7,22].

In carcasses, muscle fiber traits are associated with texture and firmness quality traits evaluated on the exposed muscle cut surface, thus influencing the sensory quality traits of meat [23]. The correlation analysis confirmed the presence of links between some traits, such as those of the TPA; indeed, the connection between hardness and chewiness and between this latter and springiness were noted. Meat color is one of the most important parameters for beef quality, and using this aspect, consumers assess its quality and decide eventually to buy it or not. The correlation analysis reported also links color parameters and the amount of lipids, as also discussed by other authors [10,24,25]. Another correlation confirmed by this analysis was the opposite trend of humidity and lipid content, which was consistent with previous studies [10,26].

The results obtained at T0 highlighted the substantial influence of the genetic type on the protein and fat content in the muscle, with Limousine as the breed with the highest protein and lower lipid percentage. The influence of the genetic type on the qualitative characteristics of meat has been reported in many studies, as well as in recent works [27,28], underlining how much genetics is gaining importance and how it is ever more difficult to separate genetics from production.

The assessment of meat quality tested with wet ageing included different aspects that might impact consumers’ perception. In our study, color traits showed a change as the wet ageing period increased; this is often characterized by a quality improvement [29].

PH values were in the normal range, but unlike what was reported by Revilla et al. (2006) [8], the variation was significant in relation to the wet ageing (T0 = 5.52 − T9/T14 = ~5.55). Under vacuum storage, there may be an increase in pH linked to a protein breakdown, combined with the formation of ammonia and amines, due to meat spoilage. However, this increase may also be associated with a natural change in pH during the first few days of storage rather than with real degradation [30].

As expected, during the storage period, there was a drop in the humidity of the samples, even though the wet ageing technique generally involves a lower loss of water, especially in the longer storage periods [31].

Meat color is an important parameter for beef quality, and in our study, these parameters fell within the normal range. The wet aged meat samples resulted in a constant increase in the lightness of the cut surface, in agreement with a previous study [9]. The L* parameter reached greater values at T14, which was what also happened in Lee et al.’s (2008) study [32]. Furthermore, b* values showed an increase from 6.41 at the beginning of wet ageing to 6.98 at the 14th day, but this result was in contrast with the study of Crivelli et al. (2019) [9], wherein significant differences of b* estimates have not been found in relation to the meat wet maturation. However, Ha et al. (2019) [22] provided very similar results to our study, suggesting that b* changes depended on many factors and were not even affected by ageing.

Hardness showed a linear pattern from 1.81 at ageing time 1 to 1.03 at time 14. Furthermore, ref. [9] found a tenderizing of the meat using the Warner Blatzer analysis technique. Field et al. (1971) [33] also found a decrease in maximal force values from 2 to 21 days post mortem.

An evident change in the sensorial quality of bone-in wet aged beef was still confirmed from the significance presented for cohesiveness, chewiness, and springiness. In accordance with Ruiz de Huidobro et al. (2003) [34], there was a decrease in chewiness and in springiness with increasing storage days.

The other fixed effects (breed, animal age, EUROP-classification, and step) led to a lower influence compared to the conservation time effect. The influence of breed on meat quality is well known, even if studies reported conflicting results depended on the parameter taken into consideration. This could be because of other influences including nutrition and management. Surely, it is evident that genetic progress has improved the quality of products such as in Limousine breed, which has been selected for meat production [35], tending towards a reduction in adipose tissue (as was also reported in this study).

EUROP classification in some studies has been considered a system based on too general indicators and that don't contemplate the complex and heterogeneous entity of a carcass [36]. The imperfect trend between the tested parameters and the EUROP-classification could be due to the lack of a correct quality assessment.

An increase in stressful conditions for the animal, due to farm management, can even lead to negative changes in meat quality, such as: increasing pH, darkening color, toughness [37]. In the present work, the increase in the events of moving animals from one farm to another could have induced stress phenomena influencing the quality, i.e., the parameters of humidity, yellow color, and meat tenderness. It was noteworthy to point out that statistical changes in the aforementioned parameters were especially reported moving from zero to one transport, suggesting that animals became used to the movements, or more likely, they suffered due to the event, independently of the number of times it occurred.

## 5. Conclusions

The data from this study indicated how the origin and post-mortem treatment of the product can affect the quality. The duration of post-mortem ageing had a positive effect on some traits of meat quality. Color and texture were improved by increasing the conservation time until 14 days of wet ageing. Due to the limited information and the biological system complexity, as well the sampling effect, more studies are needed to better investigate the effect of wet ageing and the relationship of this with other factors, such as animal welfare and breeding.

## Figures and Tables

**Figure 1 animals-13-00058-f001:**
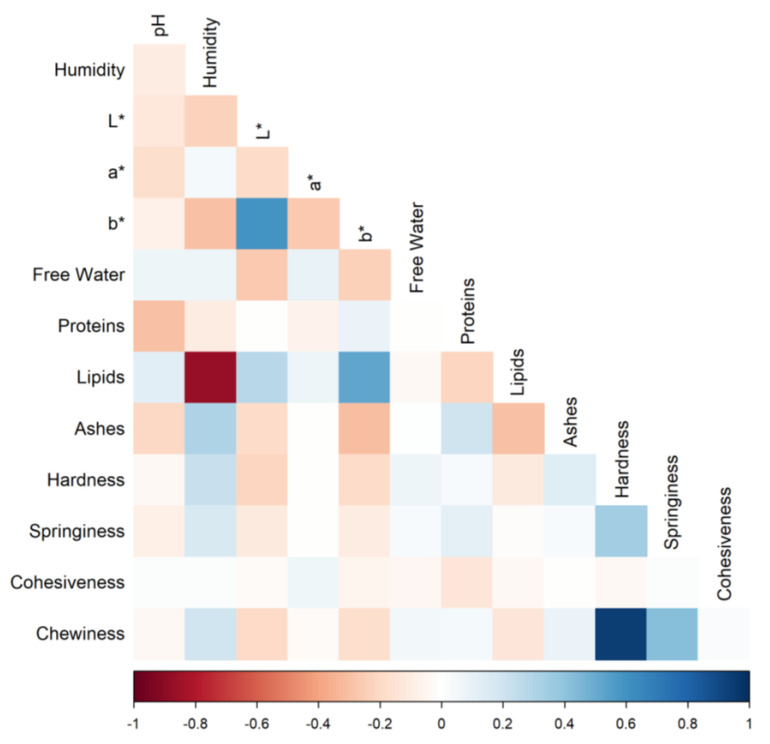
Correlation heatmap among parameters analyzed, where L* represents the lightness, a* the redness, and b* the yellowness.

**Table 1 animals-13-00058-t001:** Descriptive statistics (mean and standard deviation) of the parameters investigated, where L* represents the lightness, a* the redness, and b* the yellowness.

Parameter	Mean	sd
pH	5.66	0.09
Humidity	73.83	1.14
L*	39.93	2.03
a*	21.70	1.29
b*	5.63	1.17
Free Water	0.58	0.04
Proteins	20.94	0.58
Lipids	1.12	0.72
Ashes	1.04	0.05
Hardness	1.89	0.52
Springiness	0.90	0.02
Cohesiveness	0.42	0.04
Chewiness	0.73	0.25

**Table 2 animals-13-00058-t002:** Least squares means (SE in parentheses) of humidity, protein, intramuscular fat, and ash in samples before wet ageing.

Parameter	Humidity (%)	Protein (%)	Intramuscular Fat (%)	Ash (%)
*Breed*				
Charolais	73.40 (0.52)	20.74 ^b^ (0.39)	1.79 ^a^ (0.31)	1.02 (0.05)
Crossbred	74.13 (0.33)	21.01 ^b^ (0.26)	1.13 ^b^ (0.20)	1.06 (0.03)
Limousine	74.28 (0.36)	21.42 ^a^ (0.28)	0.86 ^c^ (0.22)	1.05 (0.04)
Romagnola	74.31 (0.40)	20.95 ^b^ (0.32)	1.00 ^b,c^ (0.25)	1.05 (0.04)
*Animal Age*				
1	73.71 (0.55)	21.44 (0.45)	1.16 (0.32)	1.00 (0.06)
2	74.18 (0.35)	20.91 (0.27)	1.09 (0.21)	1.05 (0.04)
3	74.07 (0.35)	20.95 (0.27)	1.20 (0.22)	1.06 (0.04)
4	74.15 (0.36)	20.81 (0.28)	1.33 (0.22)	1.08 (0.04)
*EUROP-conformation*				
E	74.79 (0.69)	22.14 (0.54)	0.65 (0.42)	1.04 (0.07)
U	73.67 (0.20)	21.01 (0.16)	1.29 (0.12)	1.03 (0.02)
R	73.92 (0.24)	21.02 (0.19)	1.04 (0.14)	1.06 (0.02)
O	73.72 (0.97)	19.94 (0.76)	1.80 (0.60)	1.05 (0.11)
*Step* ^1^				
0	73.62 (0.63)	21.19 ^a^ (0.49)	1.19 (0.39)	1.04 (0.07)
1	73.99 (0.36)	21.03 ^a^ (0.28)	1.36 (0.22)	1.06 (0.04)
2	74.23 (0.35)	20.63 ^b^ (0.27)	1.29 (0.21)	1.06 (0.04)
3	74.26 (0.36)	21.26 ^a^ (0.28)	0.94 (0.22)	1.02 (0.04)

^1^ Number of farms where bulls have passed through (0 = bull has remained in the same farm until transport to the slaughterhouse). ^a–c^ Statistically significant difference between groups (*p* < 0.01).

**Table 3 animals-13-00058-t003:** Least squares means (SE in parentheses) of meat parameters considering all samples, where L* represents the lightness, a* the redness, and b* the yellowness.

Parameter	pH	Humidity %	L*	a*	b*	Freewater	Hardness	Cohesiveness	Chewiness	Springiness
*Breed*										
Charolais	5.60 ^a^ (0.04)	72.98 ^b^ (0.56)	41.96 (1.21)	22.07 (0.43)	6.99 (0.57)	0.54 (0.02)	1.35 (0.18)	0.42 (0.02)	0.51 (0.07)	0.90 (0.09)
Crossbred	5.55 ^a^ (0.03)	73.65 ^b^ (0.40)	41.32 (0.84)	22.05 (0.31)	6.34 (0.41)	0.56 (0.01)	1.45 (0.13)	0.43 (0.01)	0.56 (0.05)	0.90 (0.07)
Limousine	5.52 ^b^ (0.03)	74.14 ^a^ (0.43)	41.93 (0.90)	21.90 (0.33)	6.75 (0.43)	0.55 (0.01)	1.35 (0.14)	0.44 (0.01)	0.52 (0.05)	0.90 (0.07)
Romagnola	5.48 ^b^ (0.03)	73.94 ^a,b^ (0.49)	41.13 (1.03)	22.21 (0.38)	7.04 (0.50)	0.55 (0.02)	1.28 (0.16)	0.43 (0.01)	0.49 (0.06)	0.90 (0.08)
*Wet Ageing*										
T0	5.52 ^c^ (0.03)	74.13 ^a^ (0.41)	41.09 ^c^ (0.87)	22.28 (0.33)	6.41 ^b^ ( 0.42)	0.55 ^a^ (0.01)	1.81 ^a^ (0.14)	0.40 ^d^ (0.01)	0.67 ^a^ (0.05)	0.90 (0.00)
T4	5.53 ^b c^ (0.03)	73.80 ^b^ (0.41)	41.28 ^c^ (0.87)	21.98 (0.33)	6.80 ^a^ ( 0.42)	0.56 ^a^ (0.01)	1.37 ^b^ (0.14)	0.42 ^c^ (0.01)	0.51 ^b^ (0.05)	0.89 (0.00)
T9	5.56 ^a^ (0.03)	73.48 ^c^ (0.41)	41.80 ^b^ (0.87)	22.03 (0.33)	6.95 ^a^ ( 0.42)	0.55 ^a^ (0.01)	1.21 ^c^ (0.14)	0.44 ^b^ (0.01)	0.47 ^b^ (0.05)	0.89 (0.00)
T14	5.54 ^b^ (0.03)	73.29 ^d^ (0.41)	42.18 ^a^ (0.87)	21.95 (0.33)	6.98 ^a^ ( 0.42)	0.54 ^b^ (0.01)	1.03 ^d^ (0.14)	0.46 ^a^ (0.01)	0.42 ^c^ (0.05)	0.89 (0.00)
*Animal Age*										
1	5.52 (0.04)	73.43 (0.56)	41.91 (1.71)	22.45 (0.43)	6.93 (0.56)	0.54 (0.02)	1.29 ^a,b^ (0.18)	0.44 (0.02)	0.51 ^a,b^ (0.15)	0.89 (0.01)
2	5.55 (0.03)	73.80 (0.42)	41.91 (0.88)	21.72 (0.33)	6.76 (0.42)	0.55 (0.02)	1.45 ^a^ (0.14)	0.42 (0.01)	0.55 ^a^ (0.05)	0.89 (0.00)
3	5.54 (0.03)	73.81 (0.42)	41.43 (0.88)	21.95 (0.33)	6.71 (0.43)	0.54 (0.02)	1.46 ^a^ (0.14)	0.42 (0.01)	0.56 ^a^ (0.05)	0.90 (0.00)
4	5.55 (0.03)	73.65 (0.43)	41.09 (0.91)	22.11 (0.34)	6.73 (0.43)	0.56 (0.01)	1.22 ^b^ (0.14)	0.43 (0.01)	0.46 ^b^ (0.05)	0.89 (0.00)
*EUROP-conformation*										
E	5.46 ^b^ (0.05)	74.83 ^a^ (0.84)	41.66 (1.77)	22.54 (0.66)	6.73 (0.85)	0.54 (0.03)	1.46 ^a^ (0.27)	0.44 (0.02)	0.59 (0.11)	0.91 (0.01)
U	5.62 ^a^ (0.01)	73.12 ^b^ (0.23)	42.15 (0.47)	21.51 (0.17)	7.17 (0.23)	0.57 (0.01)	1.31 ^b^ (0.07)	0.43 (0.01)	0.56 (0.03)	0.89 (0.01)
R	5.60 ^a^ (0.02)	73.59 ^a^ (0.26)	41.66 (1.77)	21.80 (0.20)	6.47 (0.26)	0.57 (0.01)	1.46 ^a^ (0.08)	0.43 (0.01)	0.50 (0.03)	0.88 (0.00)
O	5.47 ^a,b^ (0.08)	73.17 ^a,b^ (1.18)	41.29 (2.50)	22.38 (0.92)	6.77 (1.20)	0.52 (0.04)	1.20 ^b^ (0.39)	0.42 (0.03)	0.43 (0.15)	0.88 (0.00)
*Step* ^1^										
0	5.49 (0.05)	73.60 ^a,b^ (0.77)	41.30 (1.61)	21.89 (0.60)	7.24 ^a^ (0.78)	0.55 (0.02)	1.59 ^a^ (0.25)	0.43 (0.02)	0.60 (0.10)	0.89 (0.01)
1	5.54 (0.03)	73.25 ^b^ (0.41)	42.09 (0.86)	21.95 (0.32)	7.08 ^a^ (0.41)	0.55 (0.01)	1.20 ^b^ (0.13)	0.43 (0.01)	0.53 (0.05)	0.89 (0.00)
2	5.56 (0.03)	73.72 ^a^ (0.41)	42.13 (0.86)	22.14 (0.32)	6.64 ^b^ (0.41)	0.55 (0.01)	1.36 ^a^ (0.13)	0.43 (0.01)	0.50 (0.05)	0.89 (0.00)
3	5.55 (0.03)	74.13 ^a^ (0.42)	40.82 (0.89)	22.24 (0.33)	6.17 ^b^ (0.43)	0.55 (0.01)	1.28 ^a,b^ (0.14)	0.43 (0.01)	0.46 (0.05)	0.90 (0.00)

^1^ Number of farms where bulls have passed through (0 = bull has remained in the same farm until transport to the slaughterhouse). ^a–d^ Statistically significant difference between groups (*p* < 0.01).

## Data Availability

None of the data were deposited in an official repository.

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
