# Peer review of "Influence of Wet Ageing on Beef Quality Traits"

_animals, 2022, doi:10.3390/ani13010058_

Round 1

Reviewer 1 Report

Wet aging influence of beef quality traits

Overall opinion: The research presented in the manuscript is actual and very interesting. It provides information about the effect of wet aging on the quality traits of beef obtained from different cattle breeds and categories. The influence of aging on beef quality, especially sensory attributes, and technological properties, has been widely investigated, and today it is considered that aging, among other factors, significantly improves the overall quality of beef. Although the topic is not new, the results of this research can contribute to a better understanding of this topic, especially the data on the effect of the length of aging on the investigated parameters.

The authors systematically planned and conducted the research, applying an adequate methodology. The paper is well-cited, and the results and discussion appear to be correct.

However, there are some minor shortcomings that need to be addressed, before the paper is published.

Need to check the grammar throughout the manuscript.

Specific comments are in the attached manuscript.

Author Response

Authors thank reviewers for suggestions and comments. The whole manuscript has been checked and English improved. A point by point response has been done in the attached pdf.

Reviewer 2 Report

The manuscript is very interesting and well-done. Authors tested the influence of wet aging on beef quality and also tested other factors. The results are robust and enough for the conclusions. The manuscript contributes for a better understanding of the effects of wet aging that is useful for beef production chain.

My unique major suggestion is to provide a short discussion regarding the costs/benefit of this technique.

Minor remarks are described below.

Minor

Line 76, 80 and others - Longissimus thoracis using italic

Line 87 - content using a modified method of [16].  – complete this phrase

Line 92  - Final pH 24h  - use the “h” to clarify for readers

Line 114 – chewiness – Chewiness

Line 185 – remove the comma after showing

Line 194/195 – This paragraph must be combined with the next one

Line 196 – I don’t agree with “extremely”, i.e., in my country it is not

Line 209 – numerous studies must contain more references

Line 220; 227 – “other authors”/ “previous studies” must contain more references

Line 228 - The L* parameter reached greater values at T14, which was what happened also in [30] study. – improve this construction

The same for with the study of [9], (line 230, 232, 235) . Use “study of Author [number]” in manuscript

Line 235 - [9]) – review

Author Response

Authors thank reviewers for suggestions and comments. The whole manuscript has been checked and English improved.

Line 76, 80 and others - Longissimus thoracis using italic

Done

Line 87 - content using a modified method of [16].  – complete this phrase

Done

Line 92  - Final pH 24h  - use the “h” to clarify for readers

Done

Line 114 – chewiness – Chewiness

Done

Line 185 – remove the comma after showing

Done

Line 194/195 – This paragraph must be combined with the next one

Done

Line 196 – I don’t agree with “extremely”, i.e., in my country it is not

“Extremely” was removed

Line 209 – numerous studies must contain more references

“numerous” has been changed with “previous”

Line 220; 227 – “other authors”/ “previous studies” must contain more references

Done

Line 228 - The L* parameter reached greater values at T14, which was what happened also in [30] study. – improve this construction

Done

The same for with the study of [9], (line 230, 232, 235) . Use “study of Author [number]” in manuscript

Changed

Line 235 - [9]) – review

Done

Reviewer 3 Report

General comments:

the authors investigate the influence of wet aging on beef quality traits. The paper fits well with the scope of the journal.

Specific comments:

L16: please add a brief introduction and the aim of the research.

L48: the aging process is also a way to enhance the shelf life of the meat, please add this information and make focus on that, because there are several different techniques, cite: 10.3390/antiox11050827 to support this point.

L69: please report a table of the sample characteristics and distribution.

L69: a limitation of the study is the different number of samples you used, the design is unbalanced, please report as a limitation. however, I am sure that with an appropriate statistical design you can handle it.

L121: did you check for normality and homoscedasticity of your data? check as reported in 10.1080/1828051X.2020.1827990

L121: the random effect of the model, within the type and origin help to handle an unbalanced design like yours, please add.

Table 1: were interaction effects tested?

L193: the discussion needs to be increased with more comparisons and deeper analysis of the obtained results.

L261: limitations of the study need to be added.

Author Response

Authors thank reviewers for suggestions and comments. The whole manuscript has been checked and English improved.

General comments:

the authors investigate the influence of wet aging on beef quality traits. The paper fits well with the scope of the journal.

Specific comments:

L16: please add a brief introduction and the aim of the research.

Done

L48: the aging process is also a way to enhance the shelf life of the meat, please add this information and make focus on that, because there are several different techniques, cite: 10.3390/antiox11050827 to support this point.

We thank the reviewer. Surely an analysis of the shelf life will be a future test to be carried out but the current work has only examined the physical parameters of the meat, not going to evaluate the components related to the real shelf life, which is not the focus of this study

L69: please report a table of the sample characteristics and distribution.

Done. The authors think that could be boring and not necessary to include the distribution of 13 parameters.

L69: a limitation of the study is the different number of samples you used, the design is unbalanced, please report as a limitation. however, I am sure that with an appropriate statistical design you can handle it.

L121: did you check for normality and homoscedasticity of your data? check as reported in 10.1080/1828051X.2020.1827990

The normality and homoscedasticity have been checked. A sentence has been added.

L121: the random effect of the model, within the type and origin help to handle an unbalanced design like yours, please add.

Authors think that the animal random effect is necessary to correct the model because repeated measures are present for each sample, because we are analysing a sample and not the whole populations and because to not increase the breed effect. However, we think that is not necessary to explain why we included the animal as random effect because we think that it is clear.

Table 1: were interaction effects tested?

Interactions have been tested and no significance has been found

L193: the discussion needs to be increased with more comparisons and deeper analysis of the obtained results.

Some additions have been made.

 L261: limitations of the study need to be added.

Added in conclusion.

Round 2

Reviewer 3 Report

No other concerns are present with regard to the manuscript. The paper can be considered for publication on Animals in the present form.